# Sex Differences in Otolaryngology: Focus on the Emerging Role of Estrogens in Inflammatory and Pro-Resolving Responses

**DOI:** 10.3390/ijms22168768

**Published:** 2021-08-16

**Authors:** Sheng-Dean Luo, Tai-Jan Chiu, Wei-Chih Chen, Ching-Shuen Wang

**Affiliations:** 1Department of Otolaryngology, Kaohsiung Chang Gung Memorial Hospital and Chang Gung University College of Medicine, Kaohsiung 833, Taiwan; rsd0323@cgmh.org.tw (S.-D.L.); jarva@adm.cgmh.org.tw (W.-C.C.); 2Graduate Institute of Clinical Medical Sciences, College of Medicine, Chang Gung University, Taoyuan 333, Taiwan; kuerten@cgmh.org.tw; 3Department of Hematology-Oncology, Kaohsiung Chang Gung Memorial Hospital and Chang Gung University College of Medicine, Kaohsiung 833, Taiwan; 4School of Dentistry, College of Oral Medicine, Taipei Medical University, Taipei 110, Taiwan

**Keywords:** sex hormone, ENT diseases, estrogens, resolution of inflammation, resolvins, specialized lipid mediators, sex difference, otolaryngology, lipoxins, formyl peptide receptor 2

## Abstract

Otolaryngology (also known as ear, nose, and throat (ENT)) diseases can be significantly affected by the level of sex hormones, which indicates that sex differences affect the manifestation, pathophysiology, and outcomes of these diseases. Recently, increasing evidence has suggested that proinflammatory responses in ENT diseases are linked to the level of sex hormones. The sex hormone receptors are present on a wide variety of immune cells; therefore, it is evident that they play crucial roles in regulating the immune system and hence affect the disease progression of ENT diseases. In this review, we focus on how sex hormones, particularly estrogens, regulate ENT diseases, such as chronic rhinosinusitis, vocal fold polyps, thyroid cancer, Sjögren’s syndrome, and head and neck cancers, from the perspectives of inflammatory responses and specialized proresolving mediator-driven resolution. This paper aims to clarify why considering sex differences in the field of basic and medical research on otolaryngology is a key component to successful therapy for both males and females in the future.

## 1. Introduction

Ear, nose, and throat (ENT) diseases are referred to as diseases of the ear, nose, and throat, as well as those related to the head and neck structures. Examples include chronic rhinosinusitis (CRS), vocal fold polyps, hearing loss, smell and taste disorders, Sjögren’s syndrome (SS), and head and neck cancers (HNCs) [1,2,3,4,5,6,7,8,9]. ENT diseases significantly affect the everyday life of those afflicted, because they affect their ability to hear, smell, or speak properly. However, most ENT diseases are often overlooked because the patients are also found to be suffering from other comorbid conditions, such as autoimmune diseases, cardiovascular diseases, diabetes, and dementia [10,11,12,13,14,15,16,17]. These comorbidities show significant differences between males and females, implicating that sex plays an important role in the progression of ENT diseases.

Sex differences include differences in sex hormones and their effects on organ systems. In fact, sex hormones can positively or negatively affect the immune system [18,19,20,21,22,23]. Males predominantly produce testosterone from the testes, whereas small amounts of estrogens and progesterone are produced by other organs, such as the adrenal glands and peripheral tissues. In contrast, females predominantly produce estrogens and progesterone from the ovaries, whereas little testosterone is produced from the adrenal glands [23,24]. Testosterone plays important roles in regulating immune cells, such as T-helper 1 (Th1) and natural killer cells [25]. Estrogens induce T-cell homing through up-regulation of the C-C chemokine receptor type 5 expression [26]. These findings indicate that sex hormones influence the immune system and result in some immune responses being sexually dimorphic, which eventually determines the fate of the disease progression [19,20,22,23].

Most ENT diseases are associated with chronic inflammatory conditions in which the endogenous mechanism to restore tissue homeostasis is altered. Therefore, an appropriate approach to prevent or ameliorate ENT disease progression should involve the activation of the resolution pathway of the inflammatory process. The resolution of inflammation is a highly controlled and coordinated process that is mediated by specialized pro-resolving mediators (SPMs), such as resolvins and lipoxins [27,28]. SPMs are biosynthesized from ω-3 or ω-6 by 5-lipoxygenase [29,30]. These SPMs modulate immune activities, such as the infiltration of immune cells, removal of apoptotic cell debris, and inhibition of the synthesis of proinflammatory cytokines [29,30]. Recently, several studies have demonstrated the critical connection between sex hormones and specialized pro-resolving mediators, implying a significant difference between sexes in terms of disease progression and outcome [31,32,33]. For example, some studies demonstrated that SPMs, such as resolvins and lipoxins, can resolve uncontrolled inflammation and maintain tissue homeostasis in ENT diseases [34,35,36,37]. As the ENTs are chronic inflammatory diseases and their rate of occurrence and severity differs between sexes, in this review, we present the current understanding of sex differences in the relationship between the SPM-dependent resolution of inflammation and outcomes of ENT diseases. Despite the growing significance of the effect of sex hormones on the immune system in ENT disease progression, it has rarely been characterized.

Studies that investigate the effect of sex hormones, such as testosterone or progesterone, on the immune system are more limited compared to those that examine the effects of estrogens. Therefore, this paper summarizes the sex differences, relating mostly to estrogens, in ENT diseases. We also summarize the recent advances made in clarifying the effect of estrogens in some ENT diseases from the perspectives of inflammatory responses and specialized pro-resolving mediator-driven resolution (Table 1), and discuss the underlying immunological mechanisms that can regulate the progression and outcome of these diseases.

## 2. Sex Differences and the Influence of Sex Hormones in ENT Diseases: Sex Hormone Regulation of Inflammatory Responses and Their Resolution

### 2.1. Chronic Rhinosinusitis

#### 2.1.1. Effect of Sex Hormones on Inflammatory Response in Chronic Rhinosinusitis

CRS is a significant health burden; its global prevalence in the general population is estimated at 1.0–12.1% [198,199]. CRS is a chronic inflammatory disease of the nasal and sinus mucosa, and can be classified into CRS with nasal polyps (CRSwNP) and CRS without nasal polyps (CRSsNP) [4,200]. Increasing reports have suggested that the pathogenesis of these two groups is linked to sex differences. According to these studies, CRSwNP occurs more frequently in males than females, whereas CRSsNP occurs more frequently in females than males. Although men have a higher prevalence of CRSwNP, females are more likely to develop severe symptoms than males [200,201,202,203,204].

The mechanism behind these sex differences observed in CRSwNP is yet to be determined, but the findings presented below provide significant implications for understanding this mechanism. Estrogen receptors (ERα and ERβ) are expressed on immune cells and play significant roles in inflammatory responses [38,39,40]. For example, the activation of the estrogen signaling pathway in ERα-expressing eosinophils leads to the degranulation of eosinophils, which causes the release of various proinflammatory factors [41,42]. Therefore, the degranulation of eosinophils is considered a key pathogenic event in chronic eosinophilic diseases, including eosinophilic CRS. Note that the estrogen levels fluctuate during the female menstrual cycle, and this affects the number of eosinophils present in the nasal mucosa [41,43,202]. Consistent with the proinflammatory role of estrogens, the reduction of eosinophilic inflammation through tamoxifen, an estrogen antagonist, was observed in an animal asthma model [44,45]. Similarly, as high levels of estrogens were found in patients with allergic rhinitis, tamoxifen could inhibit the development of allergic reactions in the nasal cavity in a mouse study through competition with estrogens binding to the ERs [46,47]. However, regarding nasal polyps, due to the lack of relevant animal model systems, it remains unclear whether tamoxifen can reduce eosinophilic inflammation.

In clinical studies, the apoptosis of eosinophils is tightly regulated by estrogen during the female menstrual cycle. Therefore, the number of eosinophils remains constant during the menstrual cycle. In contrast, the level of estrogens significantly decreases after menopause, leading to increased numbers of eosinophils in nasal polyp samples compared to healthy donors [48,49,50]. Interestingly, these studies also suggest that ERα is also highly expressed on the surface of eosinophils [49,50], although the level of estrogens significantly decreases after menopause in patients with nasal polyps. These studies imply that estrogen therapy is useful for post-menopausal women, as the binding of estrogen to ERα was reported to play an important role in regulating the functions of eosinophils, including their mobilization and apoptosis [51]. In summary, inflammation contributes to CRSwNP; thus, a possible explanation for why females are more likely to develop severe CRSwNP symptoms than males involves enhanced inflammatory reactions driven by the levels of sex hormones, such as estrogens.

CRSwNP is also recognized as Th2-dependent eosinophilic nasal inflammation, whereas CRSsNP is known as non-eosinophilic nasal inflammation [25,52,53,200]. Estrogens are known to exert significant control on humoral immunity and increase the amount of inflammatory infiltration of eosinophils in the nasal mucosa through stimulation of Th2 cytokine production [25,52,53,200]. Females have higher levels of estrogen, which skews the immune response toward Th2, whereas males have higher levels of androgen (AR), such as testosterone, which skews the immune response toward Th1 [52,54,55]. In addition, AR signaling can increase the ratio of lung regulatory T cells (Tregs) to Th2 cells and reduce eosinophil infiltration and Th2 cytokine production in allergic airway inflammation [56,57]. Therefore, it is evident that females with CRSwNP are more likely to develop severe symptoms than males. Given that female and male hormones usually act in opposing manners, hormone-dependent mechanisms affect the regulation of the immune response under both normal and pathological conditions, leading to different disease outcomes between females and males.

#### 2.1.2. Effect of Sex Hormones on Resolution of Inflammatory Response in Chronic Rhinosinusitis

Regarding the resolution of the inflammatory response, CRSwNP is defined as an uncontrolled inflammatory disease with nose dysfunction for more than 12 weeks without any sign of resolution [4]. The persistent inflammation of the sinus epithelium, often linked to the accumulation of immune cells (i.e., eosinophils) with inflammatory signals, is intensified into a route in which the natural healing process is misoriented, eventually leading to the destruction of the nasal epithelium [53,198]. Although the underlying role of estrogens in regulating the resolution of uncontrolled inflammation in CRSwNP remains largely unknown, ER polymorphism has been implicated in female airway hypersensitivity and asthma through regulation of the activities of eosinophils, and the regulatory machinery might also play a role in resolving CRSwNP [58,59]. To resolve uncontrolled inflammatory responses, a natural preresolving mechanism that is critical in maintaining tissue homeostasis is apparently altered in CRSwNP under the influence of estrogens [60,61]. For example, a specialized proresolving lipid mediator, lipoxin A4 (LXA4), is critical for reducing innate immune cell trafficking, facilitating the resolution of acute inflammatory responses. LXA4 is an arachidonic acid (AA) metabolite generated from 12/15-lipoxygenase (12/15-LO) enzymes, which exhibit anti-inflammatory effects [62,63]. LXA4 has recently received considerable attention because of its ability to reduce inflammatory conditions and its multifaceted regulation of certain experimental disease models [36,63,64,65].

For instance, a previous study showed that epithelial cells and peripheral blood derived from aspirin-sensitive patients (higher incidence in females with CRSwNP) had significantly lower levels of LXA4 than aspirin-tolerant patients [66]. Hence, the resolution of inflammation machinery can be altered by the lower levels of LXA4, and thus, chronic inflammation is common in female aspirin-sensitive patients who also have CRSwNP. Considering the hormonal factors in this female-biased disease, these findings further indicate that hormonal status plays a role in regulating the resolution of inflammatory responses, which, if altered, can worsen the disease outcome for these patients.

Recently, many more findings on specialized proresolving mediators (SPMs) in CRSwNP and CRSsNP have been reported [36,37,61,67]. SPMs such as resolvin D2 (RvD2), derived from omega-3 fatty acid, are elevated in both CRSwNP and CRSsNP as compared to non-CRS controls [61]. In addition, LXA4 was found to be significantly increased in CRSwNP compared to CRSsNP. This study also found that an alteration of SPM pathways (e.g., RvD2, LXA4, RvD1, LTD4, LTE4, PGD2, and 11β PGF2α) was linked to CRSwNP and that an aberrant signaling of SPMs can contribute to persistent inflammation and bacterial colonization in CRS [61]. The cause of such phenomena is unknown, but we speculate that estrogens play a fundamental role. Our speculation is based on the evidence that severe symptoms occur predominantly in females, even though men have a higher prevalence of CRSwNP [68,69]. In addition, the dynamic expression of estrogens regulates SPM activity during a proper menstrual cycle, embryo implantation, pregnancy, and delivery [70,71]. Therefore, estrogens can negatively regulate SPM expression, leading to the development of severe CRSwNP in women.

### 2.2. Age-Related Hearing Loss

#### 2.2.1. Effect of Sex Hormones on Inflammatory Response in Age-Related Hearing Loss

The World Health Organization (WHO) estimates that approximately 466 million people worldwide have disabling hearing loss and that this number will increase to 900 million by 2025 [205]. Untreated hearing loss can significantly affect a patient’s daily life. For instance, hearing loss in children includes impaired or delayed language and speech development, worsened educational performance, and impaired cognitive development [72,206]. In older adults, hearing loss has been independently associated with dementia, cognitive impairment, major depressive disorder, social isolation, and increased risks of hospitalization, falls, and mortality [72,206]. Hearing loss is more than just an obstacle to communication, and its negative effects permeate and influence all aspects of the lives of those afflicted. According to the Centers for Disease Control and Prevention in the United States, “Those who have hearing loss are more likely to have low employment rates, lower worker productivity, and high healthcare costs”, indicating a significant burden of hearing loss on the healthcare system [73,74].

A growing body of evidence has suggested that sex hormones are linked to the differences in hearing loss between aging men and women [72,75,76]. The Baltimore Longitudinal Study on Aging showed that tested frequencies declined more frequently in women than in men, particularly in higher frequency regions [77]. Similar to the Baltimore aging study [77,78], another cross-sectional study also found a significant association between serum estradiol levels and hearing thresholds (as measured ranging from 0.5 to 8 kHz) in post-menopausal women and concluded that lower levels of estrogen are associated with decreased hearing sensitivity [79]. Although women tend to suffer from age-related hearing loss (ARHL), several studies have suggested that men are exposed to more damaging noise over a lifetime, which may exacerbate ARHL [72,80]. The contributing factors to ARHL in men include differences in working environments and hobbies between the sexes, and these are less likely to be dependent on changes in sex hormonal status in men than in women.

Both clinical and nonclinical studies have suggested that estrogens play a role in protection against ARHL [75,76,81]. ERα and ERβ are expressed in the auditory brainstem and can be regulated by estrogens to respond differently between post-menopausal women and control subjects [76,82]. For instance, low estrogen levels during menopause are positively associated with hearing loss [81]. In contrast, a clinical study proved that post-menopausal women receiving estrogen therapy showed protective effects against ARHL [75,81,83]. The function of estrogens in hearing protection can also be found in Turner’s syndrome [84,85]. Females with Turner’s syndrome do not produce estrogens, and some young patients often exhibit otitis media and progressive sensorineural hearing loss. In males, the prevalence and severity of ARHL is more pronounced in the absence of confounding factors, such as noise exposure [72]. In addition, males are subjected to weaker otoacoustic emissions due to the high level of androgen hormones produced in-utero during their sexual development [86,87].

The abovementioned studies clearly indicate the existence of sex differences in ARHL. Estrogens are primary hormones in females; unlike the case of CRSwNP, where estrogens elicit eosinophil infiltration, estrogen signaling can play a significant role in regulating cochlear macrophages [88,89]. Cochlear macrophages might have a resident purpose in ARHL pathogenesis because these cells express considerable ERs [31,90]. The increased recruitment of macrophages is crucial to mediate the immune response upon infection and is critical for tissue homeostasis. Regarding the ER expression on sensory cells in the cochlea, various types of ERs are expressed in the cochlear hair cells at varying levels, whereas ERβ seems to play a predominant role in the maintenance of cochlear function during aging or following acoustic trauma [91,92]. Estrogen signaling through ERβ has been shown to enhance antioxidant, antiapoptotic, and anti-inflammatory responses, all of which can contribute to hearing preservation [90,93,94]. Together, the signaling pathway activated by estrogens on different cell types can be manipulated according to different microenvironments, and such compelling results show that we are still just beginning to understand the complete complexities of sex hormones, especially estrogen regulation in sex differences.

#### 2.2.2. Effect of Sex Hormones on Resolution of Inflammatory Response in Hearing Loss

While the mechanisms of cochlear resolution remain unclear, significant efforts have been made to understand the main factors involved in the resolution of inflammation in other disease models [27,29,95]. Such studies can serve as important evidence for future studies on resolving ARHL inflammation. For example, several proresolving mediators that can promote the resolution of inflammation and tissue repair have been characterized, and a key factor identified is the glucocorticoid-inducible protein annexin A1 (ANXA1), which exerts proresolving/anti-inflammatory effects in regulating macrophage activities [84]. Other studies have demonstrated that the function and expression level of ANXA1 can be enhanced by estrogens in various experimental models [31,33,96].

Of all the studies conducted on the cochlear area so far, only one has explored the presence of resolution mediators and their receptors in the cochlea. In particular, the research group found that ANXA1 is secreted by the macrophages or Hensen cells lining the cochlea in the animal model. The results indicated that ANXA1 acts as a bridging or signaling molecule, connecting apoptotic cells to nonprofessional phagocytes and inducing the phagocytosis of apoptotic cells [97]. This signaling network can be indirectly supported by the research result that the treatment of glucocorticoids induces the release of ANXA1 from Hensen cells to strictly regulate the infiltration of leukocytes into the scala media [97]. Consequently, this process can not only promote the clearance of apoptotic cells, but also avoid uncontrolled inflammation and tissue damage caused by excessive leukocyte infiltration. Moreover, ANXA1 facilitates the clearance of apoptotic hair cells by inducing the transformation of supporting cells in the organ of Corti to nonprofessional macrophages [98,99].

The receptor for ANXA1, called ALX/FPR2, is also expressed in the scala media and cells lining the scala tympani and scala vestibule of the cochlea, which are rich in sensory outer hair, inner hair, Deiters, and pillar cells [97]. Therefore, the proresolving ANXA1 released by macrophages or Hensen cells can target the ALX/FPR2 receptor to induce the resolution of inflammation in ARHL. Increasing evidence has suggested that the ALX/FPR2 receptor is expressed on the surface of professional and nonprofessional phagocytic cells, and that its activation via the ANXA1 pathway can promote phagocytosis to clear apoptotic cellular debris [65,100,101]. In contrast, loss of ALXF/FPR2 can lead to exacerbated inflammation [102,103,104]. Together, these nonclinical results suggest that estrogens can modulate macrophage-mediated proresolving actions in the cochlear and that a proresolution therapeutic intervention by targeting macrophages can be important for ameliorating ARHL. However, there are insufficient studies on this topic in the inner ear, and thus, more studies are required.

### 2.3. Sjögren’s Syndrome

#### 2.3.1. Effect of Sex Hormones on Inflammatory Response in Sjögren’s Syndrome

SS is a chronic rheumatic disease characterized by the lymphocytic infiltration of the salivary and lachrymal glands, causing dry eyes and mouth symptoms [1,2,207,208]. Although SS is a well-known systemic disease that differs from most ENT diseases, which are localized, its geographical distribution is notable because most patients affected by SS also exhibit otorhinolaryngological manifestations [1,10,12]. For instance, patients with SS also show bilateral parotid enlargement, recurrent sinusitis, hearing loss, nasal crust formation, and dysphonia [12,13,14,209,210,211,212,213,214,215]. The incidence rate of SS is estimated to be 6.9 per 100,000 people annually, with an average age of 56 years. SS predominately occurs in females as compared to males at a 9:1 ratio, with distinct immunopathologic differences that are apparently influenced by sex hormones [2,216,217,218,219,220]. Although SS is not a life-threatening disease, its clinical manifestations increase patients’ emotional burdens [14,221,222,223,224,225]. For example, most SS patients show fatigue, poor sleep quality, and pain due to eye and mucosal dryness [13,14,222,223,224,226]. Moreover, significant lymphocytic infiltration of salivary glands causes uncontrolled inflammatory response, leading to hyposalivation, which is responsible for soreness, adherence of food to the mucosa, dysphagia, difficulties in speaking or eating, dental caries, and periodontal issues, all of which largely affect the patients’ quality of life [21,105,226,227,228,229,230,231,232,233]. As otorhinolaryngologists are often the first practitioners to notice if patients show signs of SS, an ENT examination should be performed for all patients to devise early diagnosis with prompt treatments or referral to other specialists, preventing morbidity and mortality.

The cause of SS is a multifactorial process that involves interaction between genetic (e.g., HLA, Ro52, IL-10, and TNFα) and environmental (e.g., EBV virus infection [105,106,107,108,109,110], comorbid with rheumatoid arthritis (RA), and radiotherapies) factors [14,111,112,113,114,115,116,117,118,207], which lead to aberrant inflammatory responses mediated by lymphocytes (e.g., T and B cells) and Th17 cells (e.g., IL-17) [76,77,81,82,83,84,85]. When these inflammatory responses are not resolved, they proceed to chronic inflammation, which causes significant tissue damage in the epithelium of exocrine glands [105,107,228,231]. As SS mostly affects women, the regulation of sex hormones and inflammation can provide clues for its pathogenesis [22,216,219,220]. Increasing evidence has shown that the hallmarks of SS are uncontrolled inflammation (e.g., IFNγ, IL-1β, IL-6, and IL-17) and elevation of autoantibody (i.e., Ro/SSA and La/SSB) production found in the exocrine glands, and that these pathological responses are directly influenced by sex hormones [25,119,120,121,122,123,124,125,126,127,128,129,130,131]. For example, estrogen or androgen receptors are expressed in lymphocytes and salivary gland cells [38,40,132]. The activation of ERs in B cells results in increased levels of antibodies and autoantibodies, whereas androgens decrease B-cell maturation and reduce antibody and autoantibody production [133,134,135,136]. Increased ER-β expression in the salivary acinar cells was found to be associated with SS, which is linked to higher susceptibility of apoptosis. Moreover, cytokine receptors, such as the IL-1 receptor, can be found in most hormone-producing tissues (i.e., adrenal glands, placenta, and ovaries) [137,138,139]. These results indicate a dynamic interplay between sex hormones and the immune system in a normal physiological state and disease progression.

According to the literature, estrogens perform the functions of inducing the differentiation of dendritic cells, stimulating T-cell proliferation, increasing Th2 responses, activating Treg, modulating Th17 responses, and activating M2 tissue healing-type macrophages [38,39,140,141,142]. In addition, it can inhibit innate Toll-like receptor (TLR) responses and reduce IFNγ production in immune cells [38,143]. Furthermore, estrogens can protect the disease progression of SS by reducing SS-related proinflammatory responses, such as IL-17, IFNγ, and autoantibodies Ro/SSA and La/SSB [20,82,144,145,146,147,148,216].

Some clinical studies have further suggested that the loss of estrogens (i.e., menopause) promotes SS development [146,147]. More direct evidence regarding how estrogens regulate SS can be found in some nonclinical studies. For example, a recent study on ovariectomized mice to model menopause showed significantly increased inflammation in the lacrimal and salivary glands, and estrogen replacement successfully reversed this effect by reducing the lymphocytic infiltration in the glands, suggesting that estrogens can directly reduce SS disease progression [145,149]. However, estrogens can also act as a double-sided blade in SS development. One significant example is observed in pregnant women, where an increased amount of estrogens promote the production of SS-related autoantibodies Ro/SSA and La/SSB [150,151]. 

#### 2.3.2. Effect of Sex Hormones on Resolution of Inflammatory Response in Sjögren’s Syndrome

Sustained and unresolved inflammation has now been recognized as an important feature in many autoimmune diseases, including RA, systemic lupus erythematosus, and SS [152,153,154]. This unresolved inflammation is attributed to the misregulated acute inflammatory responses, which accumulate proinflammatory signals over time, leading to chronic inflammation with an altered resolution of inflammatory responses [152,153,154]. Under normal conditions, these acute inflammatory responses are tightly controlled and should be resolved in a timely manner to avoid the occurrence of chronic status [28,30,95]; however, such a proresolving mechanism seems to be altered in SS [155]. In fact, there is a class of chemical mediators produced in our body that can be used to mitigate unresolved inflammation. These chemical mediators are now recognized as SPMs, which have been demonstrated to limit uncontrolled inflammation while promoting natural healing processes by blocking inflammatory cytokine signaling and stimulating tissue repair [27,29,30,234]. For example, one of these SPMs, called resolvin D1 (RvD1), can prevent the TNFα-mediated disruption of the salivary gland epithelium and promote the survival of salivary gland cells in nonclinical studies [35,228,235,236,237]. In particular, RvD1 can ameliorate SS-like disease progression in an SS-like mouse model by maintaining the integrity and function of the salivary gland epithelium [35,236,238]. Furthermore, treatment with AT-RvD1 can increase the saliva flow rate and inhibit the infiltration of Th17 cells in the salivary glands [35,235,236]. Note that mice lacking the RvD1 receptor ALX/FPR2 show significant impairment in both innate and adaptive immunities, along with an elevated production of proinflammatory cytokines and antibodies [103,104,236]. These results suggest that the resolution of the inflammation pathway is altered in SS, and the use of therapies such as AT-RvD1 can promote the resolution of inflammation to reverse SS progression.

As SS is predominately found in women, several nonclinical studies have also revealed that treatment with RvD1 is more effective in female mice [35,104]. Therefore, we can speculate that sex hormones are involved in the resolution of inflammation. Some experimental evidence has suggested that ovariectomized female mice (to mimic menopause) show significantly increased inflammation, and that estrogen replacement can reverse the SS-like phenotype [145,149]. These results further demonstrate that sex hormones regulate the resolution of inflammation and tissue homeostasis, especially in salivary glands. However, the underlying mechanism is still limited and requires more evidence to support this notion in a clinical setting. For instance, a recent nonclinical study suggested that sex difference is involved in regulating the resolution of inflammation using an SS-like animal model [155]. The results indicated that the RvD1 level in female mice after developing SS-like features is altered in plasma and saliva. Besides the level of lipid mediators found to be altered in SS-like female mice, the enzymes responsible for SPM synthesis, 5-LOX and 12/15-LOX, were also altered in SS-like female mice. Moreover, female mice lacking the RvD1 receptor ALX/FPR2 showed exacerbated salivary gland inflammation, reduction in saliva flow rate, and increased population of CD-20 positive B cells [104]. In contrast, ALX/FPR2 activation through RvD1 plays a critical role in promoting the resolution of inflammation and regulating antibody production in B cells. Together, these nonclinical studies further indicate that sex hormones play important roles in regulating the resolution of inflammation in SS. However, these results only indicate the potential underlying mechanisms that can elucidate sex differences in SS. Future studies should investigate the SPM level and the activity of SPM biosynthetic enzymes in SS patients.

### 2.4. Head and Neck Cancers

#### 2.4.1. Effect of Sex Hormones on Inflammatory Response in Head and Neck Cancers

Numerous epidemiological studies have indicated that sex difference is an important factor for cancer incidence and survival. In general, females have a lower risk and better prognosis than males in most cancer types, such as colon, head and neck, esophagus, lung, and liver [239,240]. The higher incidence in the male population can be attributed to diet, occupational exposure (i.e., chemicals or carcinogens), and unhealthy lifestyle (i.e., smoking and alcohol consumption). HNCs (including nasopharyngeal carcinoma and oral squamous cell carcinoma) are examples that display sex difference with a higher incidence rate in males than females at a 5:1 ratio [241,242,243]. HNCs are the seventh most common tumors worldwide, with an estimated annual burden of 625,173 new cases and 323,160 deaths in 2018. More than 90% of all HNCs are oral squamous cell carcinoma, arising in the mucosal surface lining the aerodigestive tract [243]. According to the WHO, the incidence rates of HNCs are high in certain Asian-Pacific countries [8,241]. Although their prevalence is low in Western countries, the incidence rates of oral cancer driven by human papillomavirus infections and certain lifestyle habits have increased significantly and become a public health concern [9,244]. Besides the multiple factors (i.e., alcohol and smoking) that lead to HNC development, sex hormones can also contribute to head and neck carcinogenesis [156,157]. For instance, previous clinical studies on HNC patients have shown that estrogen levels in females play a protective role in developing cancer [158,159,160,161,162]. As males have lower levels of estrogen, they are more predisposed to develop cancer. Moreover, the destruction of liver function in alcoholics leads to an alteration in the metabolism of estrogens. Several studies have shown that alcohol can interfere with the balance of sex hormones by promoting the aromatization of androgens into estrogens and thereby impairing the ratio of androgens to estrogen [163,164,165,166].

As estrogens and their receptors have been found to be present in oral cavity, laryngeal, or hypopharyngeal cancers, we can speculate that estrogens play a direct role in regulating HNC progression [141,156,157]. For instance, recent studies have suggested that an altered expression of ERs is found in the malignant tissues of oral mucosa and is correlated with HNC survival [156,167]. Moreover, higher estrogen levels are linked with lower HNC risk in women who have undergone hormone replacement therapy, are pregnant, or given birth below 35 years of age [158]. In contrast, higher HNC risk is associated with menopause onset before 52 years of age [158]. Although estrogens have been proven to play a protective role in some studies, other studies have shown opposite results [168,169]. The mechanisms underlying these contradictory results originate from the alteration and fluctuation of endogenous estrogen levels in HNC progression [156,157]. In particular, a previous study showed that 75% of young women who never smoked and drank were HNC patients at age 19–39 years [170], implying that estrogens at different physiological statuses may affect HNC tumorigenesis, probably due to the polymorphism of estrogen and its receptors in the expression level.

Furthermore, increasing evidence has suggested that estrogens can regulate a wide variety of cellular functions, such as anti-inflammation, proliferation, migration, and differentiation of cancer cells and other cells within the tumor microenvironment [31,38,119,168,171]. Among the cells involved in the inflammatory tumor microenvironment, the essential player, the macrophage, is involved during all phases of inflammation and can be regulated by estrogens as it expresses ERs [31]. When the immune system fails to resolve inflammation, the chronic inflammatory response dominates and influences important metabolic functions, including cell homeostasis and genomic changes, which eventually lead to the development of carcinogenesis [172,173,174,175]. The infiltrated tumor-associated macrophages are major inflammatory cells that promote the progression of malignancies by supporting tumor growth and shaping the tumor microenvironment by secreting proinflammatory cytokines, including the TNF superfamily, interleukins (i.e., IL-1, IL-8, and IL-6), chemokines (i.e., CXCL-10), prostaglandins, and reactive oxygen species (ROS), which promote the development of oral cancers [175,176]. Specifically, TNF-α, IL-6, and PGE-2 participate in the process of submucous fibrosis, which plays a critical role in enhancing the malignant transformation of oral cancer cells [177,178]. Several nonclinical studies have shown that circulating estrogens can regulate the activities of macrophages and reduce inflammatory responses. Specifically, ER activation by estrogens can decrease the synthesis of proinflammatory factors (e.g., TNFα, IL-6, and COX-2) in macrophages to clear the damaged proteins through the activation of proteasomes [179,180]. Moreover, estrogens inactivate NF-κB-mediated inflammatory responses via the pathway involved in the activation of the estrogen-activated receptor function and phosphatidylinositol 3-kinase [179]. Recent studies have also suggested that estrogens decrease the expression of TLR4, which plays a critical role in producing proinflammatory cytokines [143,181]. Therefore, these results suggest that the regulatory function of macrophages can be modulated by estrogen and linked to the reduction of proinflammatory responses that limit the growth of tumors.

However, some opposite results regarding the effect of estrogens in cancers have also been reported. For instance, some experimental studies have suggested that estrogens exhibit genotoxic, mutagenic, and carcinogenic effects. In particular, a previous study suggested that genes or proteins associated with estrogen metabolism (i.e., ERα, Erβ, and the androgen receptor) are highly expressed in isolated human head and neck cells and that these molecules can contribute to the tumorigenesis of HNC [157,182,183].

In addition to estrogens, other sex hormones, such as androgens, can also contribute to the tumorigenesis of HNC. It has been demonstrated that AR is expressed in patients with oral squamous cell carcinoma (OSCC), and the expression of this receptor is critical for promoting OSCC growth [184,185]. A more recent study using clinical samples found that more than 20% of neoplastic OSSC epithelium patient samples were stained positive with androgen receptors [184]. Considering that ERs and ARs play roles in the tumorigenesis of HNC, the aforementioned studies identify some issues that are worth further investigation, including whether the ratios of the expression levels of estrogen and androgen and their receptors in cancer cells play roles in the tumorigenesis of HNC.

#### 2.4.2. Effect of Sex Hormones on Resolution of Inflammatory Response in Head and Neck Cancers

While the cause of HNC has been associated with the abovementioned risk factors, unresolved chronic inflammation has been implicated as a critical driving force that can lead to genetic and epigenetic changes in HNC malignancies [173,178,186,187]. For example, the pathogenesis of oral squamous cancer cells involves several oral inflammatory conditions, such as oral submucous fibrosis, oral lichen planus, discoid lupus erythematosus, oral ulcers related to repetitive tissue injury, and chronic periodontal disease [188,189]. The infiltration of inflammatory cells in oral submucous fibrosis mainly comprises lymphocytes, plasma cells, and macrophages, which are altered by tumor cells to produce proinflammatory cytokines (i.e., TNF-α, IL-8, and IL-6), prostaglandins, COX-2, and ROS, elevating the inflammatory status in the local microenvironment, which favors tumor growth [186,187,190]. Therefore, targeting these tumor-associated inflammatory factors appears to be a tempting strategy to devise novel therapeutics for treating oral cancers.

Furthermore, recent studies have explained the application of SPMs that can resolve uncontrolled inflammation and maintain tissue homeostasis. These SPMs are derived from ω-3 or ω-6 polyunsaturated fatty acids through enzymatic processing with lipoxygenases (i.e., 5- and 12/15-LO) [32,191,192,193]. For example, resolvins and lipoxins are SPMs that can be naturally synthesized from AA, docosahexaenoic acid and aspirin, and perform potent proresolving and anti-inflammatory functions by regulating the infiltration of immune cells, removing apoptotic cell debris, inhibiting the synthesis of proinflammatory cytokines (e.g., TNF-α, IL-6, CXCL10, COX-2, and MCP-1), and inhibiting tumor migration/metastasis [32,191,192,193,194]. These findings suggest the proresolving actions of SPMs in anticancer functions. Although the involvement of sex hormones in regulating the biological events between SPMs and HNC remains unclear, a recent study has suggested a potential link of how estrogens regulate the resolution of inflammation in HNC. In particular, a proresolving mediator, LXA4, has been found to share structural similarities with estrogens (i.e., estrogen 17-estradiol) and compete with ER to inhibit estrogen’s function, which indicates the therapeutic potential of LXA4 in treating estrogen-associated diseases, such as cancers [64,195]. Moreover, LXA4 or another proresolving mediator RvD1 can inhibit estrogen-induced epithelial–mesenchymal transition through ALX/FPR2 receptor signaling in several disease models, including endometriosis, lung cancer, and nasopharyngeal carcinoma [31,34,100,103,155,177,193,196,197]. Together, these results further imply an important link between estrogens and the resolution of inflammation in HNC; however, the exact mechanism remains largely unknown. 

## 3. Conclusions

This review article aims to highlight the interconnections among sex hormones, ENT diseases, and immune responses. We discussed several ENT diseases that involve both sex hormones and immune responses, and found that sex hormones play important roles in regulating inflammatory responses and resolution of inflammation after tissue injury. In addition, we reviewed the recent findings that have shown that the machinery in immune response is mostly altered in ENT diseases under the influence of hormonal status (Table 1).

To date, studies addressing the role of estrogens in ENT diseases have largely been restrained by significant experimental obstacles. For example, most in vitro settings conducted using exogenous estrogens or antagonists to ERs are too simplified to address the complex in vivo system. Moreover, the dose, duration, and route of estrogen treatment in both in vitro and in vivo models should be considered as important factors in future experimental designs. Estrogen signaling can exert completely opposite effects on ENT diseases for improvement or augmentation, depending on the microenvironment and cell type.

In conclusion, estrogens and other sex hormones play important roles in regulating the immune system and thus influence the disease outcomes. Hence, increased efforts are required to identify their functions in terms of driving the inflammation and its resolution in ENT diseases. This is particularly important given that the established importance of sex differences in ENT disease can yield a novel therapeutic strategy to ameliorate or reverse ENT disease progression.

## Figures and Tables

**Table 1 ijms-22-08768-t001:** Sex difference in ENT diseases from the perspectives of inflammatory responses and specialized proresolving mediator-driven resolution.

ENT Diseases	Sex Bias	Level of Estrogens	Sex Hormone Receptors	Cells Involved in the Pathogenesis	Cytokines Involved in the Pathogenesis	Involvement of Resolution of Inflammation Factors	References
Chronic rhinosinusitis with nasal polyps	1. Male 2. Female with lower estrogens	Low estrogens	High ERα	1. Th2, eosinophile 2. Epithelial cells 3. Macrophages	IL-4, IL-5, IL-13, IL-25, IL-33	1. Intake of omega-3 fatty acid show delaying incidence of recurrence 2. Alteration of RvD2, LXA4, RvD1, LTD4, LTE4, PGD2, and 11β PGF2α profile	[38,39,40,41,42,43,44,45,46,47,48,49,50,51,52,53,54,55,56,57,58,59,60,61,62,63,64,65,66,67,68,69,70,71]
Chronic rhinosinusitis without nasal polyps	Female	N/A	N/A	1. Th1 cells 2. Neutrophil 3. DC	IFN, IL-6, IL-8, IL-17, TGFβ	1. Alteration of PGD2 and 2. TXA2 profile	
Age-related hearing loss	1. Male 2. Females with Turner’s syndrome	Low estrogens	ERα/ERβ	1. Cochlear Macrophages 2. Cochlear hair cells 3. Hensen cells	TNF-α, IL-1β, IL-6, IL-8	ANXA1	[72,73,74,75,76,77,78,79,80,81,82,83,84,85,86,87,88,89,90,91,92,93,94,95,96,97,98,99,100,101,102,103,104]
Sjögren’s syndrome	Female	Low estrogens (during menopause)	High ERβ	1. Lymphocytes (T and B cells) 2. Th17 cells	TNF-α, IL-1β, IL-6, IL-17, IFNγ	RvD1	[105,106,107,108,109,110,111,112,113,114,115,116,117,118,119,120,121,122,123,124,125,126,127,128,129,130,131,132,133,134,135,136,137,138,139,140,141,142,143,144,145,146,147,148,149,150,151,152,153,154,155]
Head and Neck Cancers	Male	Low estrogens	High ERα/ERβ/AR	1. Macrophages 2. Stromal cells	TNF-α, IL-6, CXCL10, COX-2, IL-1a, IL-1b, IL-4, IL-8, and TGFb	1.LXA4 2. Resolvins (RvD2, RvD4 and RvD5)	[156,157,158,159,160,161,162,163,164,165,166,167,168,169,170,171,172,173,174,175,176,177,178,179,180,181,182,183,184,185,186,187,188,189,190,191,192,193,194,195,196,197]

## Data Availability

No new data were created or analyzed in this study. Data sharing is not applicable to this article. Data available in a publicly accessible repository.

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
