# Peer review of "Sex Differences in Otolaryngology: Focus on the Emerging Role of Estrogens in Inflammatory and Pro-Resolving Responses"

_ijms, 2021, doi:10.3390/ijms22168768_

Round 1
Reviewer 1 Report
The authors provide an in-depth thoughtful review on the importance of sex differences in Ear, Nose and Throat diseases. The connection between immune responses influenced by sex hormones is evident and significant. This article is very helpful to highlight the need for increased sex-based research in many fields that have fallen behind, including Ear Nose and Throat diseases. However, the clarity in how the authors try to provide evidence is lacking. The authors are encouraged to edit this paper diligently for grammatical errors and word choice to help with clarity.
The authors should clarify/use whole words for titles of sections rather than using abbreviations. i.e. 2.1 CRS, 2.3 SS, 2.4 HNC. Also need to define all acronyms before using them throughout the article.
What effect does the fluctuating estrogen levels throughout menstruation have on CRS? Is it the estrogen receptors playing a major role or is it influenced by circulating estrogen? Which receptor is it, alpha, beta or both? Does testosterone have any effect on CRSwNP or CRSsNP? Can this section be clearer?
Explaining/defining “proresolving mechanisms” earlier in the article will provide the reader with more clarity when explaining different metabolites that trigger this mechanism.
- See Basil and Levy., 2016 DOI: 10.1038/nri.2015.4
Lines 160-164, it is stated that the BL Study on Aging found tested frequencies declined in women more frequently than men, but then it is followed up with men having more exposure to more damaging noises. Wouldn’t that seem to point to men having a greater decline in tested frequencies? If that is the case, this section feels a bit confusing on who is at a greater risk.
Authors briefly mentioned the contradictory response of estrogen in some head and neck cancers. Can also look at the Women’s Health Initiative for more data/studies on estrogen replacement therapy, although not specifically geared towards head and neck cancer.
Author Response
We would like to thank you for your valuable comments and effort to improve this manuscript. We have addressed all comments as can be seen in the attached file. The original reviewer comments are shown in italics, while our responses are highlighted in red.

Reviewer 2 Report
Although the article talks about sex differences in otolaryngological diseases, it doesn't take in account psycological differences which might affect the disease as well. The article has many flaws, too many references, and some really important were not mentioned. It has also lot of language mistakes so it needs careful reading and rewriting. The results should be easily visible. The scope of the article is too wide, maybe the article should be focused on one field or disease.
Author Response

(The authors gave the same response as above.)

Reviewer 3 Report
Chapetrs can not start with abbreviation such as 2.3. . SS and 2.4. . HNC
Alcohol consumption promotes aromatization of androgens to estrogens and this fact should be added.
Despite siginifacance of estrogen, testosterone and androgen receptor are shown to be significant in H&N cancer and this should be at least mentioned in order to avoid opinion that only estrogen and ER are key players. Actually, the most important is ratio between E and T in ENT diseases
Tomasovic-Loncaric C, Fucic A, Andabak A, Andabak M, Ceppi M, Bruzzone M, Vrdoljak D, Vucicevic-Boras V. Androgen Receptor as a Biomarker of Oral Squamous Cell Carcinoma Progression Risk. Anticancer Res. 2019 Aug;39(8):4285-4289.
Batelja-Vuletic L, Tomasovic-Loncaric C, Ceppi M, Bruzzone M, Fucic A, Krstanac K, Boras Vucicevic V. Comparison of Androgen Receptor, VEGF, HIF-1, Ki67 and MMP9 Expression between Non-Metastatic and Metastatic Stages in Stromal and Tumor Cells of Oral Squamous Cell Carcinoma. Life (Basel). 2021 Apr 10;11(4):336. doi: 10.3390/life11040336.
Author Response
We would like to thank you for your valuable comments and effort to improve this manuscript. We have addressed all comments as can be in the attached file. The original reviewer comments are shown in italics, while our responses are highlighted in red.
